# Missingness Bias Calibration in Feature Attribution Explanations

**Shailesh Sridhar**
University of Pennsylvania
shai2403@seas.upenn.edu

**Anton Xue**
University of Texas at Austin
anton.xue@utexas.edu

**Eric Wong**
University of Pennsylvania
exwong@seas.upenn.edu

## Abstract

Popular explanation methods often produce unreliable feature importance scores due to missingness bias, a systematic distortion that arises when models are probed with ablated, out-of-distribution inputs. Existing solutions treat this as a deep representational flaw that requires expensive retraining or architectural modifications. In this work, we challenge this assumption and show that missingness bias can be effectively treated as a superficial artifact of the model's output space. We introduce MCal, a lightweight post-hoc method that corrects this bias by fine-tuning a simple linear head on the outputs of a frozen base model. Surprisingly, we find this simple correction consistently reduces missingness bias and is competitive with, or even outperforms, prior heavyweight approaches across diverse medical benchmarks spanning vision, language, and tabular domains.

## 1 Introduction

As black-box deep learning systems are increasingly deployed in high-stakes settings such as medicine, finance, and law, there is increasing demand for reliable and trustworthy model explanations. A common approach for explaining model predictions is to use feature attribution methods, which assign importance scores to input features based on their influence on the output. Popular methods, such as LIME (Ribeiro et al., 2016) and SHAP (Lundberg & Lee, 2017), estimate these scores by perturbing the input, typically by ablating selected features and measuring the change in prediction. Because true feature removal is often infeasible (e.g., one cannot physically delete image pixels or omit words from tokenized sequences), attribution methods approximate removal by substituting the selected features with default or placeholder values, such as black pixels or special tokens (Ancona et al., 2017; Sundararajan et al., 2017).

These substitutions often result in out-of-distribution inputs that deviate significantly from the model's training data, inducing a systematic distortion in predictions known as *missingness bias* (Hase et al., 2021; Hooker et al., 2019; Jain et al., 2022). Such bias can severely undermine the reliability of explanations. As illustrated in Figure 1, a classifier that accurately detects a brain tumor from clean inputs fails to do so when irrelevant features are masked, demonstrating how seemingly innocuous ablations can corrupt model behavior. Since perturbation-based attributions are derived directly from these corrupted predictions, their reliability is fundamentally compromised, leading to inconsistent feature importance scores (Duan et al., 2024; Goldwasser & Hooker, 2024; Hooker et al., 2019). This also opens the door to adversarial manipulation: malicious actors can exploit this vulnerability to design deceptive models that obscure their use of sensitive attributes such as race or gender (Joe et al., 2022; Koyuncu et al., 2024; Slack et al., 2020).

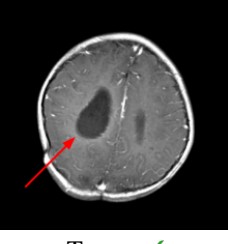 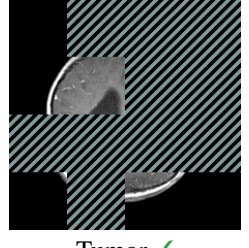 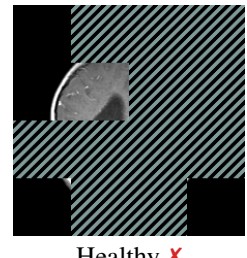

Tumor ✓      Tumor ✓      Healthy ✗

Figure 1: **Removing irrelevant features can cause a misdiagnosis.** A fine-tuned ViT (Dosovitskiy et al., 2020) correctly predicts "tumor" on the clean image (left) and a subset of the relevant features (middle). However, masking irrelevant features flips the prediction to "healthy", despite the tumor remaining visible (right). For visualization, gray stripes denote zero-valued pixels, and images are contrast-boosted.

A variety of mitigation strategies have been proposed to address missingness bias. *Replacement-based* methods aim to reduce distributional shift by imputing masked features with more realistic content (Agarwal & Nguyen, 2020; Chang et al., 2018; Kim et al., 2020; Sturmfels et al., 2020). *Training-based* methods fine-tune or retrain the model to better handle ablations (Hase et al., 2021; Hooker et al., 2019; Park et al., 2024; Rong et al., 2022), while *architecture-based* approaches embed robustness directly into the model via structural design changes (Balasubramanian & Feizi, 2023; Jain et al., 2022).

However, these strategies are often impractical. Replacement-based methods are usually specialized to specific domains (e.g., text (Kim et al., 2020)) or might require training model-specific imputations (Chang et al., 2018). On the other hand, training-based solutions require intensive engineering and computing resources, while architecture-based modifications require a deep understanding of model internals. Moreover, it is also increasingly common that models are complete black boxes, such as when interacting with API-based LLM providers.

In this work, we question whether such complex interventions are necessary. We investigate a simple yet surprisingly powerful strategy for mitigating missingness bias: finetuning a linear head on the outputs of a frozen base model. This approach, which we call **MCal**, is *lightweight*, *model-agnostic*, and *post-hoc*. It is significantly cheaper in implementation effort than training-based methods, does not require model-specific adaptations like architecture-based and replacement-based methods, and only needs access to the model's output logits. In the following, we summarize the development of MCal and our contributions.

**A New Perspective on Missingness Bias.** We find that missingness bias, a problem often treated as a deep representational flaw, can be effectively mitigated with a simple post-hoc correction in the model's output space. This finding suggests the bias is often a superficial artifact, challenging the prevailing assumption that expensive retraining or architectural modifications are necessary.

**A Lightweight Method with Theoretical Guarantees.** We formalize this approach as MCal, a lightweight calibrator that is highly efficient to optimize (Section 3). Furthermore, our simple formulation provides theoretical guarantees of convergence to a globally optimal solution, ensuring a level of stability and reproducibility rare for deep learning interventions.

**A Strong and Practical Baseline.** We demonstrate MCal's effectiveness across diverse models and data modalities, where it is often competitive with heavyweight approaches (Section 4). This establishes a strong and practical baseline that can be immediately adopted by researchers and practitioners to improve the reliability of their explanations.

## 2 UNDERSTANDING MISSINGNESS BIAS

Perturbation-based feature attribution methods like LIME (Ribeiro et al., 2016) and SHAP (Lundberg & Lee, 2017) evaluate models on inputs with ablated features, typically replaced by fixed

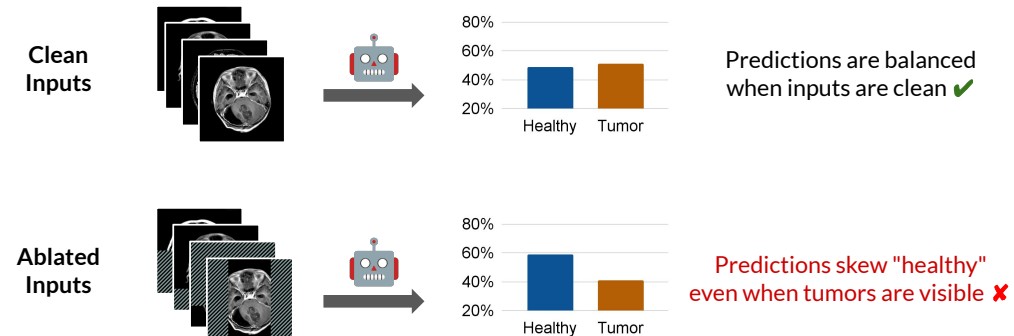

Figure 2: **Feature ablations induce class distribution shifts.** Masking non-critical regions skews predictions towards the "healthy" class, even when tumors remain visible. This effect, known as *missingness bias*, causes the model to misclassify inputs that retain relevant features, and undermines the reliability of feature attribution explanations.

baseline values (e.g., zero-vectors or mean-pixel values). However, because these synthetic inputs often fall outside the model's training distribution, they can induce systematic prediction distortions, a phenomenon known as *missingness bias*. This section provides a background on this bias and its consequences for explanation reliability.

## 2.1 PATHOLOGY: SYMPTOMS AND MEASUREMENTS

The effects of missingness bias are not merely statistical curiosities; they manifest as tangible failures that undermine the reliability of explanation methods.

**Systematic Skew in Predictions.** The most direct failure mode of missingness bias is a systematic skew in model predictions (Jain et al., 2022). As illustrated in Figure 2, the model's accuracy degradation is not random but systematic: it develops a consistent bias towards one class (in this case, "healthy") even when the core evidence for the correct class remains visible. This failure mode is particularly pernicious, persisting even when we selectively avoid masking the central image patches most likely to contain the tumor.

**Unreliable Feature Attributions.** Another consequence of this degraded accuracy is that any feature attributions derived from the model are fundamentally unreliable. If a model's predictions are incorrect on ablated inputs, the importance scores computed from these predictions cannot be trusted to reflect the model's true reasoning. Empirical findings support this; for instance, Jain et al. (2022) show that feature importance scores from models with high missingness bias fail standard robustness tests such as top-k removal. Prior work has also shown that minor changes to the ablation process can yield vastly different explanations, suggesting they reflect perturbation artifacts rather than genuine model logic (Hooker et al., 2019).

**Quantifying Missingness Bias.** Many feature attribution methods operate under the assumption that feature ablation is a neutral act of intervention intended to simulate the removal of information (Sturmfels et al., 2020; Sundararajan et al., 2017). When a model's behavior deviates from this expected neutrality, the resulting shift in its aggregate predictive distribution serves as a direct measure of missingness bias. This shift is typically quantified as the distribution shift between the class frequencies on the clean data distribution $\mathcal{D}$ versus the ablated data distribution $\mathcal{D}'$ (Balasubramanian & Feizi, 2023; Jain et al., 2022):

$$\mathsf{MissingnessBias}(f) = D_{\mathrm{KL}}\Big(\mathop{\mathbb{E}}_{x' \sim \mathcal{D}'} \mathsf{Class}(f(x')) \,\big\|\, \mathop{\mathbb{E}}_{x \sim \mathcal{D}} \mathsf{Class}(f(x))\Big), \tag{1}$$

where $\mathcal{D}'$ is the distribution of inputs where each feature is i.i.d. ablated with some given probability, and let $\mathsf{Class}(f(x))$ be the one-hot vector representation of the class predicted by $f$ on $x$. The above can then be understood as a measure of information-theoretic "surprise" when $f$ is evaluated on *unbiased* ablations, supposing only knowledge of its behavior on clean inputs. In particular, Jain et al.

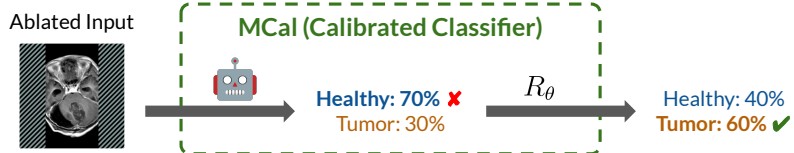

Figure 3: **MCal corrects class distribution shifts induced by input ablations.** The model initially predicts "healthy" from the ablated input. MCal applies a learned transformation $R_\theta$ to adjust the output probabilities, thereby restoring alignment with expected class distributions. This calibration method is model-agnostic, requiring only the classifier's output probabilities of each class.

(2022) specifically introduces this to measure missingness bias, rather than of adjacent phenomena, such as prediction sensitivity with respect to top-$k$ feature selections (Hase et al., 2021).

## 2.2 THE CHALLENGE OF MITIGATION

A variety of strategies have been proposed to address missingness bias, which can be broadly categorized as follows:

- *Replacement-based.* These methods aim to make ablated inputs appear more in-distribution. Beyond simple values (e.g., zero and mean-valued (Hase et al., 2021)), more complex variants include marginalization, which averages outputs over plausible replacement values (Chirkova et al., 2023; Frye et al., 2020; Haug et al., 2021; Kim et al., 2020; Vo et al., 2024), random noising (Rong et al., 2022), and generative modeling, which uses a secondary model to in-paint realistic content (Agarwal & Nguyen, 2020; Chang et al., 2018). However, these approaches are often complex and can introduce their own artifacts.

- *Training-based.* This approach treats feature ablations as a form of data augmentation. Methods like ROAR (Hooker et al., 2019) and GOAR (Park et al., 2024) retrain or fine-tune the model on masked inputs to align its train and test distributions. Although effective at building robust representations, this strategy is computationally expensive and only possible when the model can be modified.

- *Architecture-based.* These methods embed robustness directly into the model's design. For example, modified vision transformers (Dosovitskiy et al., 2020; Jain et al., 2022) and CNNs (Balasubramanian & Feizi, 2023) can be altered to use dedicated mask tokens or explicitly suppress the influence of ablated regions. However, these changes are often non-trivial, architecture-specific, and not generalizable.

While often effective, the high cost and complexity of these methods make them impractical for many modern use cases, especially those involving large-scale, pre-trained foundation models. Furthermore, such approaches are entirely infeasible when working with API-based models that do not permit retraining or architectural changes. This gap highlights the need for a practical, lightweight, and model-agnostic approach to mitigating missingness bias that we introduce next.

## 3 MCAL: A LIGHTWEIGHT CALIBRATOR FOR MISSINGNESS BIAS

Having established the pathology of missingness bias and the practical limitations of existing heavyweight solutions, we now introduce our method. We propose **MCal**, a lightweight, post-hoc correction that is surprisingly effective at mitigating missingness bias.

### 3.1 ARCHITECTURE AND OPTIMIZATION

The calibration process is illustrated in Figure 3. A base classifier $f : \mathbb{R}^n \to \mathbb{R}^m$ first processes an input $x$ to output the *raw logits* $z = f(x)$. A calibrator $R_\theta : \mathbb{R}^m \to \mathbb{R}^m$ then transforms the raw logits into the *calibrated logits* $R_\theta(z)$. Specifically, we implement this as an affine transform:

$$R_\theta(z) = Wz + b, \tag{2}$$

where the calibrator is parametrized by $\theta = (W, b)$, with $W \in \mathbb{R}^{m \times m}$ and $b \in \mathbb{R}^m$. To fit the calibrator, we use a standard cross-entropy objective that aligns the calibrated prediction on an ablated input with the base model's prediction on the clean input:

$$\mathcal{L}(\theta) = \mathop{\mathbb{E}}_{(x, x') \sim \mathcal{D}} \text{CrossEntropy}[R_\theta(f(x')), \text{Class}(f(x))], \quad (3)$$

where $(x, x') \sim \mathcal{D}$ are samples of a clean input $x$ and its ablated version $x'$, and $\text{Class}(f(x))$ denotes the one-hot prediction on the clean input.

Our approach is deliberately minimalist, prioritizing efficiency without compromising performance. We apply a standard cross-entropy objective, identical to that used in heavyweight retraining methods (Hooker et al., 2019), but only to a lightweight matrix-scaling calibrator (Guo et al., 2017). This design is highly efficient, with orders of magnitude fewer parameters ($m^2 + m$) than fine-tuning or even parameter-efficient methods like LoRA (Hu et al., 2022). Our experiments in Section 4 confirm that this minimalist approach is, in fact, sufficient to yield competitive performance with more engineering-intensive approaches like retraining the model or architecture modifications. Furthermore, this simple design also comes with strong theoretical guarantees on its optimization process, which we detail next.

## 3.2 Theoretical Guarantees and Geometric Interpretation

Our affine parametrization of $R_\theta$ means that standard gradient-based optimization will provably converge to an optimal solution, which we formalize as follows.

**Theorem 3.1** (Guaranteed Optimal Convergence). *The MCal objective $\mathcal{L}(\theta)$ is convex in $\theta$.*

*Proof.* The function $\mathcal{L}(\theta)$ is convex in $\theta$, as it is a composition of the convex cross-entropy loss and an affine transformation. Because local minimums are also global minimums for convex functions, standard gradient-based optimization (e.g., SGD, Adam) will converge to an optimal solution. □

The importance of this guarantee is twofold. First, it ensures reproducibility and stability: the optimization process is guaranteed to converge to the same optimal solution, reducing the need for extensive hyperparameter sweeps or random seed searches. Second, it provides a strong assurance of quality, guaranteeing that the resulting calibrator is a globally optimal affine correction for the given data.

**Geometric Interpretation.** MCal also has a clear geometric interpretation, visualized in Figure 4. The uncalibrated outputs form biased point clouds on the probability simplex, with the Class A cluster pulled towards the Class B vertex, leading to systematic misclassification. MCal learns an optimal affine transformation in the logit space that rotates, scales, and shifts these distributions. This untangles the clouds and pushes them towards their correct vertices. Theorem 3.1 guarantees that this correction is globally optimal for our parametrization.

## 3.3 Implementation Considerations

**Conditioning on Ablation Rates.** Our experience shows that the severity of missingness bias is strongly correlated with the fraction of features that are ablated. To account for this, we recommend using an "ensemble" of specialized calibrators, each one fit for a specific ablation rate (e.g., 10%, 20%, etc.). At inference time, we apply the calibrator that was trained for the ablation rate closest to that of the input. We study the advantage of this ensemble in Section 4.

**Integration with Explainers.** As a post-hoc wrapper, MCal is compatible with any perturbation-based explanation method. The calibrated model, $\tilde{f}$, can be used as a drop-in replacement for the original model, $f$, in any existing explainability pipeline. The resulting feature attributions are then generated from a model that has been explicitly corrected for the missingness bias induced by the explanation method's own perturbation strategy.

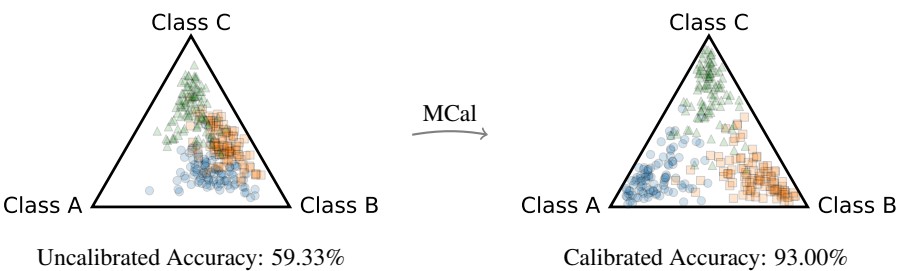

Figure 4: **Geometric intuition of MCal on a synthetic dataset.** Missingness bias causes the uncalibrated outputs to shift. For instance, the Class A cluster (blue circles) is pulled towards the Class B vertex, leading to systematic misclassification and low accuracy. MCal applies an optimal affine transformation to the uncalibrated outputs, correcting the shift and improving accuracy.

**Training Set Size and Overfitting.**  Dense parametrizations of $W$ risk overfitting when the number of parameters exceeds the number of training samples (Guo et al., 2017), which can occur when there are many classes. In such cases, the training loss may go to zero while test performance does not improve. We recommend two strategies to mitigate overfitting. First, one may consider adding a regularization term to the objective. Second, one may also consider sparse parametrizations, such as taking $W$ to be a diagonal matrix (also known as "vector-scaling"), which would reduce the total parameter count to $O(m)$.

## 4 EXPERIMENTS

We now present experiments to validate the impact of missingness bias in explainability, as well as the ability of MCal to mitigate it. Moreover, we demonstrate that MCal repeatedly outperforms more expensive baselines, such as full retraining and architecture modifications. Additional details are given in Appendix A.

**Models, Datasets, and Compute.**  We evaluate on a diverse set of medical benchmarks that span vision (Brain MRI (Nickparvar, 2021), Chest X-ray (CheXpert) (Irvin et al., 2019), and Breast Cancer Histopathology (BreakHis) (Spanhol et al., 2015)), language (MedQA (Jin et al., 2021), MedM-CQA (Pal et al., 2022)), and tabular domains (PhysioNet (Haug et al., 2021), Breast Cancer (Wolberg et al., 1993), Cardiotocography (CTG) (Campos & Bernardes, 2000)). We respectively evaluate on these domains with ViT-B16 (Dosovitskiy et al., 2020), Llama-3.1-8B-Instruct (AI@Meta, 2024), and XGBoost (Chen & Guestrin, 2016), which are trained using standard methods. For compute, we had access to a machine with four NVIDIA H100 NVL GPUs.

**Input Ablations and Calibration.**  We say that an input $x \in \mathbb{R}^n$ has ablation rate $p = k/n$ if $k$ of its features are ablated. To evaluate on a tractable range of $p$, we use $p \in \{0/16, 1/16, \dots, 15/16\}$ for vision, $p \in \{0/10, 1/10, \dots, 9/10\}$ for language, and $p \in \{0/10, 1/10, \dots, 9/10\}$ for tabular, where recall that we recover the clean input at $p = 0$. For imputations, we use zero-valued (black) patches for vision, we replace whitespace-separated words with the special string UNKWORDS for language, and we perform mean imputation for tabular data. For vision specifically, we select $k$ patches to ablate, regardless of their original values (e.g., some MRI images already have black patches). Following discussion from Section 3.3, the *unconditioned* calibrator was fit on inputs where each feature was uniformly ablated with probability $1/2$, whereas the *conditioned* ensemble has a calibrator fit at each value of $p$. All calibrators were optimized using Adam (Kingma & Ba, 2014) with a learning rate of $10^{-3}$ for 5000 steps.

**Question 1: Do calibrated models lead to better explanations?**  Missingness bias is known to skew the explanation quality of feature attribution methods (Jain et al., 2022). To that end, we consider how two representative methods, LIME (Ribeiro et al., 2016) and SHAP (Lundberg & Lee, 2017), perform on calibrated vs. uncalibrated models. These methods output a ranking of each in-

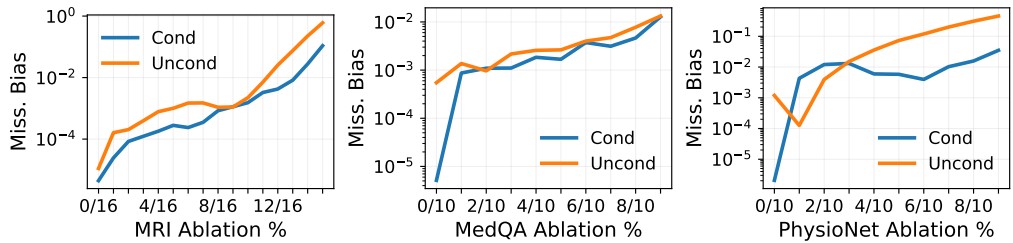

Figure 5: **Calibrated models have better explanations.** Compared to an uncalibrated baseline model (Base), LIME and SHAP explanations on MCal-calibrated models have more accurate feature importance scores (sufficiency ↓). In addition, calibrated models are also more robust to feature ablations (sensitivity ↓). Results are shown for the MRI dataset using an unconditioned calibrator.

Figure 6: **Conditioning on ablation rate improves MCal.** Fitting an ensemble of calibrators at a discretized set of ablation rates can help reduce the overall missingness rate, compared to using a single unconditioned calibrator. (Left) MRI, (Middle) MedQA, (Right) PhysioNet.

put feature's importance to the model, which we evaluate using the standard *sufficiency* metric (Hase et al., 2021), detailed in Appendix A.1. Informally, sufficiency measures whether the features identified as important are enough on their own to maintain the model's original prediction confidence (lower values indicate a higher quality ranking). We report results in Figure 5.

**Question 2: How does calibration affect model robustness?**    It is generally desirable for models to be robust to feature perturbations, as this can improve generalization and reduce the risk of adversarial behaviors. To that end, we measure the robustness of the underlying model to the removal (ablation) of features via the *sensitivity* metric, detailed in Figure 5. We show our results in Appendix A.1, which shows that the model is not overly dependent on its top-k features for prediction.

**Question 3: What is the impact of conditioning on feature ablation fractions?**    Rather than fitting a single calibrator, we observe that using an ensemble of calibrators, each conditioned upon a single fraction (ablation rate), can improve performance. We compare the performance of this conditioning in Figure 6, where we observe an improvement in performance over an unconditioned calibrator. This is expected, as a model's missingness bias is known to vary with the ablation rate (Hooker et al., 2019; Jain et al., 2022), and an ensemble thereby allows each calibrator to specialize to their respective rates.

**Question 4: How does MCal compare to the baselines?**    We compare MCal to each of the following prior approaches, which have all been employed in previous work to combat the problem of out-of-distribution inputs.

- **Base**: This is the unmodified, uncalibrated classifier that acts as a reference baseline.
- **Replacement-based (Replace)**: Our implementation of replacement-based mitigation is inspired from Hase et al. (2021). In particular, for vision, we use the channel-wise mean pixel value of the clean dataset (Carter et al., 2021). For language, we drop tokens from the sequence so that the ablated token sequence is shorter in length than the clean one (Hase et al., 2021). For tabular, we perform mean imputation.

|  | Dataset | Base | Replace | Retrain | Arch | TempCal | PlattCal | MCal (✓) |
|---|---|---|---|---|---|---|---|---|
| **Vision** | Brain MRI | $1.18\,\mathrm{e}{-1}$ | $1.51\,\mathrm{e}{-1}$ | $\mathbf{6.70\,e{-4}}$ | $1.40\,\mathrm{e}{-1}$ | $1.16\,\mathrm{e}{-1}$ | $1.27\,\mathrm{e}{-1}$ | $7.43\,\mathrm{e}{-3}$ |
|  | CheXpert | $1.70\,\mathrm{e}{-1}$ | $9.70\,\mathrm{e}{-2}$ | $2.67\,\mathrm{e}{-2}$ | $1.50\,\mathrm{e}{-1}$ | $1.65\,\mathrm{e}{-1}$ | $2.02\,\mathrm{e}{-1}$ | $\mathbf{8.82\,e{-3}}$ |
|  | BreakHis | $1.87\,\mathrm{e}{-1}$ | $4.20\,\mathrm{e}{-1}$ | $2.19\,\mathrm{e}{-2}$ | $1.54\,\mathrm{e}{-1}$ | $1.86\,\mathrm{e}{-1}$ | $1.66\,\mathrm{e}{-1}$ | $\mathbf{4.29\,e{-3}}$ |
| **Language** | MedQA | $1.61\,\mathrm{e}{-1}$ | $1.50\,\mathrm{e}{-1}$ | $1.70\,\mathrm{e}{-1}$ | $2.68\,\mathrm{e}{-2}$ | $1.57\,\mathrm{e}{-1}$ | $9.48\,\mathrm{e}{-2}$ | $\mathbf{9.44\,e{-4}}$ |
|  | MedMCQA | $1.89\,\mathrm{e}{-1}$ | $2.59\,\mathrm{e}{-1}$ | $1.52\,\mathrm{e}{-1}$ | $1.40\,\mathrm{e}{-1}$ | $7.81\,\mathrm{e}{-1}$ | $1.13\,\mathrm{e}{-1}$ | $\mathbf{9.01\,e{-3}}$ |
| **Tabular** | PhysioNet | $1.17\,\mathrm{e}{-1}$ | $1.20\,\mathrm{e}{-1}$ | $5.59\,\mathrm{e}{-3}$ | $8.14\,\mathrm{e}{-2}$ | $1.17\,\mathrm{e}{-1}$ | $1.19\,\mathrm{e}{-1}$ | $\mathbf{5.01\,e{-3}}$ |
|  | Breast Cancer | $1.02\,\mathrm{e}{-1}$ | $1.44\,\mathrm{e}{-1}$ | $5.68\,\mathrm{e}{-3}$ | $2.13\,\mathrm{e}{-1}$ | $1.02\,\mathrm{e}{-1}$ | $1.08\,\mathrm{e}{-1}$ | $\mathbf{1.92\,e{-5}}$ |
|  | CTG | $1.06\,\mathrm{e}{-1}$ | $7.02\,\mathrm{e}{-2}$ | $6.61\,\mathrm{e}{-3}$ | $2.85\,\mathrm{e}{-1}$ | $1.06\,\mathrm{e}{-1}$ | $9.20\,\mathrm{e}{-2}$ | $\mathbf{3.35\,e{-3}}$ |

Table 1: **MCal is an effective and lightweight way to reduce missingness bias.** It repeatedly outperforms more computationally expensive baselines, such as retraining and architecture modification. We report the KL divergence-based metric in Equation (1).

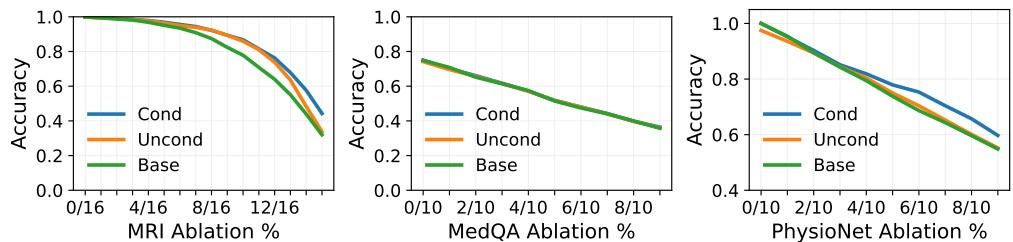

Figure 7: **MCal does not harm classifier accuracy.** Across different ablation levels, the accuracy of uncalibrated vs. calibrated classifiers is comparable. This also holds when the input is clean (ablation fraction zero). (Left) MRI, (Middle) MedQA, (Right) PhysioNet.

- **Training-based approaches (Retrain)**: Models are fine-tuned on ablated inputs, where each feature (patch, token) is uniformly ablated with probability $1/2$.

- **Architectural-based (Arch)**: We perform a non-trivial modification of ViT to accept attention masks as in Jain et al. (2022). For models with architectural support for missing features, we use those: e.g., attention masking in Llama-3 and native support for `NaN` in XGBoost.

- **Standard calibration (TempCal, PlattCal)**: We additionally consider existing calibration-based methods from literature, particularly temperature (TempCal) and vector-scaling Platt calibration (PlattCal), as described in Guo et al. (2017).

We report in Table 1 the average of values from the ensemble of conditioned calibrators. We found that MCal is often superior even to more computationally and engineering-intensive baselines, such as model retraining and ViT architecture modifications. In support of our earlier claims, we also observe that MCal outperforms both temperature and Platt calibration. Replacement-based methods have inconsistent performance, which aligns with known observations on their sensitivity to imputation values. Finally, we note that architecture-native support for missing features may, in fact, exacerbate missingness bias, as seen in XGBoost on the Breast Cancer and CTG datasets.

**Question 5: How does MCal affect classifier accuracy?** MCal fundamentally alters a pretrained base classifier $f$ into $\tilde{f}$, which is then deployed to downstream applications. Importantly, the accuracy of $\tilde{f}$ must remain high, even when it is optimized on ablated images Equation (3). We show in Figure 7 that this is indeed the case: we compare the uncalibrated base model against both ablation rate-conditioned and unconditioned calibrators. We observe that both forms of calibration improve classifier accuracy at all ablation rates, where we recall that the clean image is obtained at an ablation rate of zero. Aligning with earlier findings, we see that the conditioned calibrator outperforms the unconditioned calibrator.

## 5 RELATED WORK

**Missingness Bias in Explainability.** Missingness bias (Jain et al., 2022) denotes the systematic distortions that arise when attribution methods "remove" features via ablations, e.g., with black pixels, zero-valued embeddings, or special `[MASK]` tokens. Such ablated inputs are often out-of-distribution with respect to the model's training distribution, which can result in erratic predictions, inflated confidences, and unstable feature importance scores (Hooker et al., 2019; Vo et al., 2024). In particular, importance scores can vary drastically with the chosen replacement technique (Haug et al., 2021; Sturmfels et al., 2020) and can even be exploited adversarially (Slack et al., 2020). Consequently, feature-based explanations commonly reflect ablation artifacts rather than genuine model reasoning (Hase et al., 2021), which risks eroding trust in high-stakes settings. In addition to the methods described earlier in Section 2.2, there are several benchmarks related to missingness bias (Duan et al., 2024; Hesse et al., 2023; Liu et al., 2021).

**Calibration Methods.** A calibration method post-hoc rescales the logits or probabilities of a model prediction without modifying the underlying model weights. Classic techniques include binning (Zadrozny & Elkan, 2001), Platt scaling (Platt et al., 1999), and temperature scaling (Guo et al., 2017). This is often used to improve and calibrate model predictions under input distribution shift, such as in autonomous driving (Tomani et al., 2021), healthcare (Shashikumar et al., 2023), and LLMs (Kumar et al., 2022). A closely related work is Decker et al. (2025), which addresses perturbation-induced miscalibration but focuses on confidence rather than the systematic class-skew.

**Robust and Reliable Explanations.** There is much interest in the development of robust explanations for machine learning models. Notable efforts include the development of benchmarks for explanations, particularly feature attribution methods (Adebayo et al., 2018; 2022; Agarwal et al., 2022; Dinu et al., 2020; Duan et al., 2024; Havaldar et al., 2025; Jin et al., 2024; Kindermans et al., 2019; Nauta et al., 2023; Rong et al., 2022; Zhou et al., 2022). There is also interest in formally certifying explanations (Bassan & Katz, 2023; Bassan et al., 2025a;b; Jin et al., 2025; Lin et al., 2023; Xue et al., 2023; You et al., 2025). Other efforts, such as this work, involve adapting classifiers to be more robust to input ablations in feature attributions.

## 6 DISCUSSION, FUTURE DIRECTIONS, AND CONCLUSION

**Calibration Design.** While other calibrator parametrizations are viable, any non-convex parametrization of the objective risks losing guarantees of optimality convergence. In turn, this risks introducing undesirable behavior, such as sensitivity to the initialization of calibrator parameters. Additionally, observe that the measure of missingness bias (Equation (1)) is different than the calibrator optimization objective. This is because the missingness bias measure is not differentiable due to the one-hot Class function, which motivated us to search for reasonable alternatives, e.g., the standard cross-entropy objective in classification. While it would be interesting to explore, for instance, differentiable relaxations of Equation (1), we leave this to future work.

**Missingness Bias and Data Variability.** Visual medical datasets, e.g., Chest X-rays, often exhibit lower variability than image datasets like ImageNet (Deng et al., 2009). Despite this, missingness bias can persist in models trained on datasets such as ImageNet (Jain et al., 2022). However, it is not known how missingness bias varies as a function of both dataset variability and model architecture.

**Beyond Explainability.** Missingness bias is a fundamental risk when evaluating feature subsets on a model that is not explicitly designed to handle missing data. While we are primarily motivated by challenges in explainability, this work has broader applications. In vision, model evaluation with masked images is a standard practice. In language modeling, a token's embedding is often dependent on its position, meaning that ablations are position-sensitive, whether via the attention mask, subsetting the input sequence, or replacement with special `[MASK]` tokens.

**Limitations.** MCal requires access to a collection of clean and ablated prediction logits, which may not always be available, such as for some API-based LLMs. Even then, gradient-based optimization is only guaranteed to converge to global optimality under certain parameterizations of the

calibrator. Overfitting is also a potential risk, particularly in settings with a large number of possible classes (e.g., a language model's vocabulary size), in which case regularization is warranted. Furthermore, MCal is only intended to mitigate missingness bias, and other forms of bias in the model and data may still be propagated. While our experiments show that linear corrections on the logits suffice to mitigate missingness bias, and hence our use of the description "superficial", it may be the case that for missingness bias in certain model classes, it is harder to mitigate in this manner.

**Future Work.** One direction is to investigate the theoretical guarantees and empirical performance of different calibrator parametrizations, such as a one-layer feedforward network instead of an affine transform. Another extension is to broaden our study on the performance of calibrated classifiers in explainability, such as with respect to the explanation methods and metrics surveyed in Section 5. It would be interesting to explore methods for mitigating missingness bias when prediction logits are not available, a common restriction for API-based large language models. Additionally, the idea of calibration may also be extended to other instances of domain shift and out-of-distribution inputs, which are prevalent throughout machine learning literature.

**Conclusion.** Missingness bias threatens the reliability of popular explanation methods and techniques, a problem magnified by the increasing impracticality of existing engineering-intensive solutions. To overcome this, we introduce MCal, a lightweight calibration method that requires only a collection of clean and ablated prediction pairs. We demonstrate that a simple, affine parametrization of the calibrator offers strong theoretical guarantees while achieving empirical performance that often outperforms more expensive baselines. In summary, MCal is an efficient, model-agnostic calibration scheme that improves the reliability of popular feature-based explanation methods.

**Ethics Statement.** This work presents a method for improving the reliability of feature-based explanation methods. Our intended audience includes researchers and practitioners interested in explainability. While there may be potential for misuse, we do not believe that the contents of this paper warrant concern.

**Reproducibility Statement.** All code and experiments for this paper are available at:

https://github.com/ShaileshSridhar2403/MCal

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

# A  ADDITIONAL EXPERIMENTS AND DETAILS

We present our experimental setup here, along with any additional experiments and relevant details.

**Compute.**  We had access to a server with four NVIDIA H100 NVL GPUs.

## A.1  METRICS FOR FEATURE ATTRIBUTIONS

Given an input $x \in \mathbb{R}^n$ and a classifier $f : \mathbb{R}^n \to \mathbb{R}^m$, a feature attribution explanation method returns a vector $\alpha \in \mathbb{R}^n$ where $\alpha_i$ denotes the importance of feature $x_i$. We now discuss some metrics for evaluating the quality of the feature attribution $\alpha$.

**Sufficiency.**  One way to assess the quality of an attribution $\alpha$ is by using only its top-k-selected features for prediction. Let $\mathsf{Top}_k(x, \alpha) \in \mathbb{R}^n$ be the version of $x$ where its top-k features are selected, as ranked by $\alpha$. Equivalently, $\mathsf{Top}_k(x, \alpha)$ is the version of $x$ where its bottom $n - k$ features are ablated. Then, the top-k sufficiency metric (Hase et al., 2021) is defined as:

$$\mathsf{Sufficiency}(f, x, k) = f(x)_{\hat{y}} - f(\mathsf{Top}_k(x, \alpha))_{\hat{y}}, \tag{4}$$

where $\hat{y} = \arg\max_y f(x)_y$ is the predicted class. In this formulation, a lower sufficiency is preferable, as it indicates that the selected features can more reliably attain the confidence associated with a clean input's prediction.

**Sensitivity.**  Conversely, one can also assess how much *omitting* the top-k features would affect prediction. Analogously to the sufficiency metric, let $\mathsf{Bot}_{n-k}(x, \alpha)$ denote the version of $x$ where the bottom $n - k$ features are selected; i.e., the top $k$ features are ablated. The top-k sensitivity metric, also called *comprehensiveness* (Hase et al., 2021), is defined as:

$$\mathsf{Sensitivity}(f, x, k) = f(x)_{\hat{y}} - f(\mathsf{Bot}_{n-k}(x, \alpha))_{\hat{y}} \tag{5}$$

A higher score indicates the features were critical to the prediction, whereas a lower score suggests that the model is more robust (i.e., less sensitive) to their inclusion.

## A.2  MCAL TRAINING DYNAMICS

Here, we investigate the training dynamics and performance of MCal as the training set size varies. We show the results in Figure 8. In general, as the training dataset size increases, test-time accuracy increases. When $n$ is small, however, the problem is over-parametrized, meaning that the training loss continues to decrease without significantly improving test-time accuracy.

## A.3  MCAL VS. RETRAIN

We note an interesting phenomenon in Table 1, where MCal often outperforms retrain-based methods. At first glance, this is surprising because the model's representational capacity should exceed that of MCal. We suspect that this may be due to our use of *conditioned* MCal to train an ensemble of calibrator heads, which may at times be sufficient to overcome this gap in representational power. Indeed, we observe that *unconditioned* MCal, in which only a single calibrator head is used across all missingness levels, typically underperforms retraining.

In attempting to diagnose this observation with more intensive training runs (25K steps, $10^{-3}$ learning rate and 50 ablations per data point) we observe that while retrain can outperform MCal in certain situations, its unconditioned variant may also do so.

We suspect this may be due to the differences between the formulations of missingness bias in Equation (1) and the optimization objective in Equation (3). A more in-depth analysis would be warranted in future work. We show these trends in Figure 9, with a focus on tabular models given their ease of retraining.

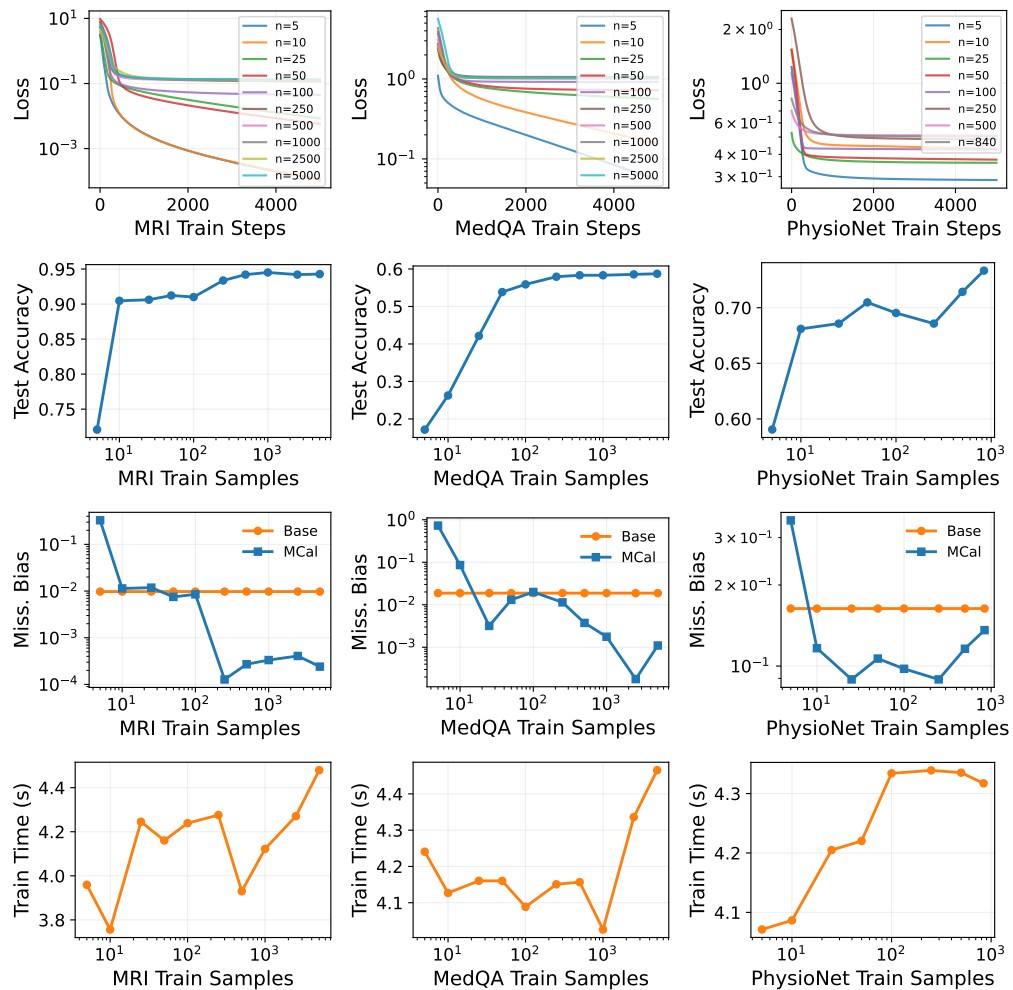

Figure 8: **MCal training dynamics and performance on different dataset sizes.** For different amounts ($n$) of clean-ablated input pairs, we show the training loss curves and test-set accuracies. Each training run consisted of $5000$ iterations, and all runs finished in $\leq 5$ seconds.

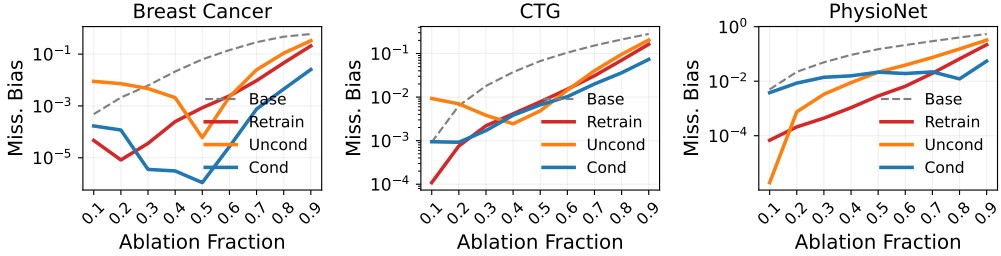

Figure 9: **MCal can outperform retrain due to its specialization** Fitting an ensemble of calibrators at a discretized set of ablation rates can even outperform retraining the model at times (Left) Breast Cancer, (Middle) CTG, (Right) PhysioNet.

## A.4 CASE STUDY: INTEGRATION WITH API-BASED MODELS

Current-day machine learning workflows are increasingly dependent on closed-weight API-based models. Recognizing this, we perform a case study demonstrating how MCal can be extended to such settings, assuming only access to output logits.

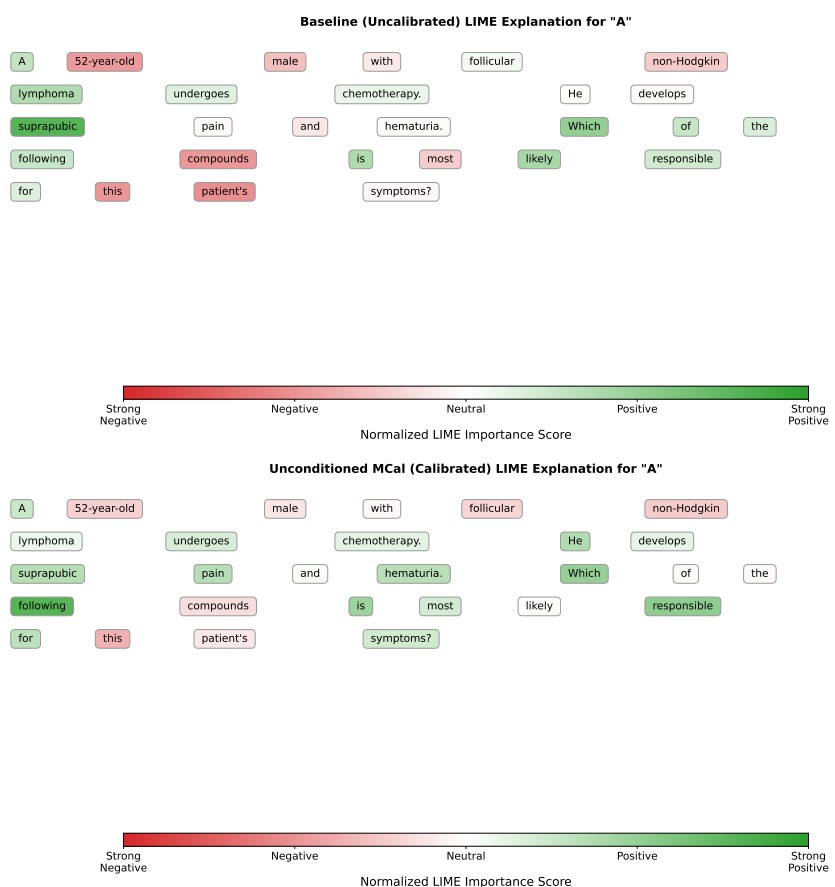

Figure 10: **For a selected example question, MCal results in different feature importances**, for example medically relevant features/terms such as "hematuria" gain importance in the calibrated heatmap. The model task is, for the above question, to choose between the Options: A: Cyclophosphamide, B: Cisplatin, C: Mesna, D: Bleomycin.

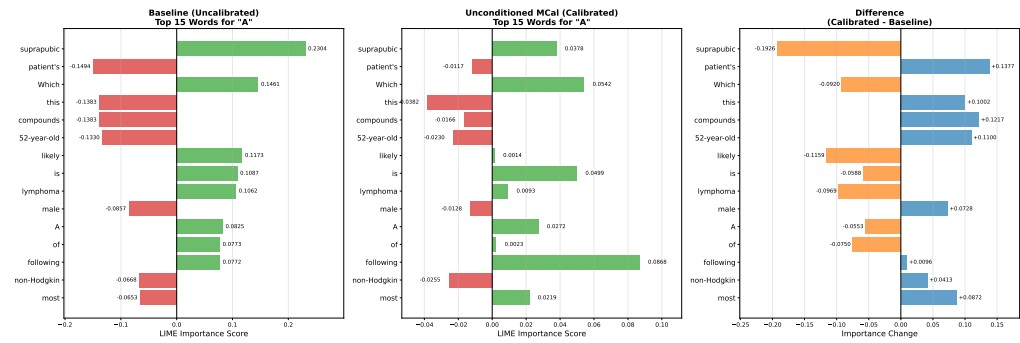

Figure 11: Direct comparison of LIME-derived feature importance values, with (a) Uncalibrated, (b) Calibrated, and (c) demonstrating the difference between the two, i.e., (b) - (a).

In particular, we demonstrate how GPT-4o-mini (Hurst et al., 2024) can be calibrated on selected MedQA instances in Figure 10 and Figure 11. The calibrated model redistributes feature importance in potentially meaningful ways, elevating diagnostic symptoms like "hematuria"(a hallmark sign of cyclophosphamide toxicity) and "pain", while reducing dominance of the anatomical term "suprapubic". This rebalancing suggests calibration may produce more clinically-aligned and faithful explanations, though domain expert validation and further faithfulness testing are needed.

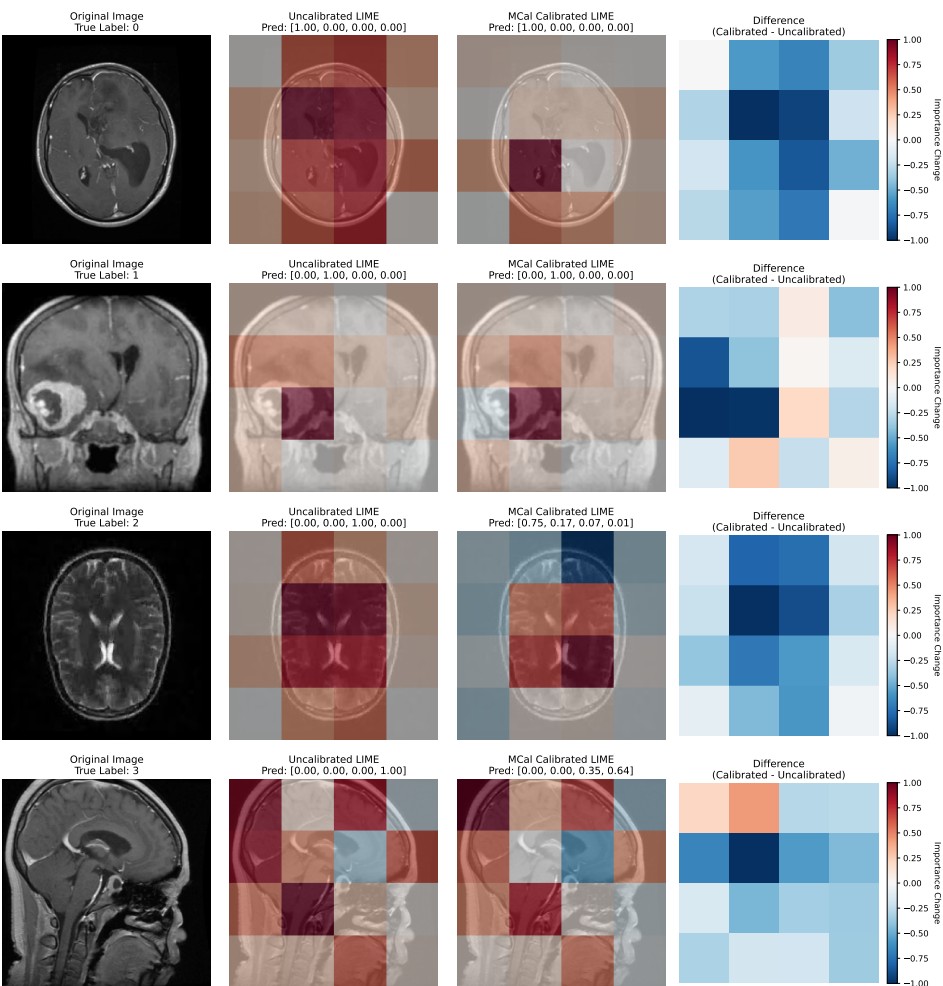

Figure 12: **Selected examples of LIME on the MRI dataset.** In calibrated models, we observe that LIME tends to assign less importance to border patches, where relevant features are less likely to occur. The four classes are: Meningioma (0), Glioma (1), Pituitary Tumor (2), and No Tumor (3).

