# OpenReview forum: "Missingness Bias Calibration in Feature Attribution Explanations"
_ICLR.cc/2026/Conference — ICLR 2026 Poster_

### Official Review · Reviewer_3ZQX · 2025-10-18

**Soundness:** 4
**Presentation:** 3
**Contribution:** 2
**Rating:** 6
**Confidence:** 4

**Summary:**

The paper presents MCal, a method that addresses the missingness or out-of-distribution (OOD) bias inherent in perturbation-based XAI methods. Unlike alternative approaches, MCal does not require computationally expensive retraining or architectural modifications. The paper demonstrates quantitatively that MCal produces improved explanations, provide comparative analysis against baseline methods, and verify that the approach preserves overall model accuracy.

**Strengths:**

- The proposed solution is elegantly simple yet well-motivated and thoroughly evaluated, though this simplicity could raise questions about the scope of the contribution.
- The method exhibits modality- and architecture-agnostic properties, being largely model-agnostic depending if the logins can be accessed.
- The manuscript is very well-structured and clearly written, facilitating reader comprehension through effective organization.
- The paper provides a pip-installable code repository, enhancing practical accessibility and reproducibility.

**Weaknesses:**

**W1:** Medical images, particularly in radiology and histopathology, exhibit relatively low inter-patient visual variability, making OOD examples straightforward to generate. But how is this for natural images? As these are more diverse, I suspect less sensitivity to such missingness biases through the perturbations.

**W2:** The evaluation relies exclusively on two metrics that are known to potentially introduce bias. The addition of qualitative visualizations comparing saliency maps before and after debiasing would significantly enhance reader understanding and provide practical insight into the method's real-world performance.

**W3:** Given the method's claimed model-agnostic nature, evaluation or at minimum code implementation with closed-source API-based models would strengthen the work. Such demonstrations, contingent on API access to model logits, would underscore the method's relevance for the broader research community that relies predominantly on closed-source models.

**Questions:**

See Weaknesses W1 - W3.

Comment: An error.txt file remains in the code repository (maintaining anonymity).

---

> ### Author Response · Authors · 2025-11-28
>
> We thank the reviewer for their patience and positive assessment, as well as their suggestions for improving practical impact. We respond to their points below.
>
>
> **W1:** The reviewer raises a good point about the relatively lower variability of many visual medical datasets. In fact, missingness bias is also prevalent in “natural” image datasets such as ImageNet [1]. However, it remains an open question how missingness bias might vary as a function of both model architecture and dataset variability. We have added a discussion of this.
>
>
> **W2:** We have incorporated this suggestion into Appendix B, where we show visualizations of LIME scores on calibrated and uncalibrated models for the MRI datasets. In calibrated models in these examples, LIME appears to assign lower scores to border regions that also coincide with lower tumor incidence. This suggests that calibrated models tend to place greater importance (as measured by LIME) on central areas, where tumors are more likely to occur.
>
>
> **W3:** This is a great suggestion. We have implemented a case study in Appendix A3 on how MCal may be applied to API-based LLMs. There, we consider GPT-4o-mini (logits API) on the MedQA dataset with LIME explanations, and find that the calibrated LLM places greater emphasis on clinically relevant terminology and reduces reliance on any one single word in the prompt. We plan to include this in our next open-sourced code release as a Jupyter notebook tutorial so that it will be accessible to the broader research community.
>
>
> [1] Jain et al. “Missingness Bias in Model Debugging”, ICLR 2022.

---

### Official Review · Reviewer_ojnE · 2025-10-30

**Soundness:** 3
**Presentation:** 3
**Contribution:** 3
**Rating:** 6
**Confidence:** 4

**Summary:**

This paper introduces a novel method called MCal, which aims to mitigate "missingness bias" in model explanations. Missingness bias occurs when input features are ablated or removed in explanation methods, leading to biased predictions that undermine the reliability of feature importance scores. The authors propose MCal, a lightweight, post-hoc calibration technique that corrects missingness bias by fine-tuning a linear head on the outputs of a frozen base model. The method is model-agnostic, cost-effective, and offers strong theoretical guarantees. The results demonstrate that MCal outperforms more complex, computationally expensive methods across a variety of domains, including medical image classification and natural language processing tasks.

**Strengths:**

1. Practicality and Efficiency: MCal is lightweight and does not require extensive computational resources, making it more accessible than existing solutions that require retraining or architectural adjustments. The use of a linear transformation and a cross-entropy loss function ensures minimal overhead.
2. Theoretical Guarantees: The paper offers strong theoretical guarantees of convergence for the proposed method, ensuring that the calibrator will consistently find an optimal solution. This gives confidence in the reproducibility and stability of the approach.
3. Empirical Validation: The authors provide extensive experimental results demonstrating MCal's effectiveness across diverse medical domains (vision, language, and tabular). It outperforms other methods such as retraining and architecture-based solutions, making it a compelling baseline for addressing missingness bias.

**Weaknesses:**

1. **Lack of Real-World Application Scenarios**: While the method is demonstrated on several benchmarks, the paper lacks concrete real-world use cases where MCal is applied to solve specific problems. Providing experimental results from a more tangible application would strengthen the paper’s argument and demonstrate the method's practical value in more complex scenarios.


2. **Lack of Discussion on the Training Process**: The authors mention that the calibration process requires training a new parameter. However, if the training dataset is too small, the model may overfit, while larger datasets may increase computational costs. The paper could benefit from a discussion and experimental analysis of how the choice of training data size affects the performance of MCal and its scalability.

**Questions:**

1. Could you provide a detailed case study or experiment in a real-world application, such as medical diagnostics where MCal has been implemented to solve a specific problem?

2. In situations with large class spaces (e.g., language models), how do you prevent overfitting during calibration?

3. How does the size of the training dataset affect the performance of MCal? What was the specific training time of your experiments? How do you balance the data size and the computational cost?

---

> ### Author Response · Authors · 2025-11-28
>
> We thank the reviewer for their patience and positive assessment and suggestions regarding the case studies. We address the main points below.
>
>
> **W1 + Q1:** This is a great suggestion. As a step towards this, we have added a case study in Appendix A3 on how MCal may be applied to API-based models and feature attributions. In particular, we consider GPT-4o-mini (logits API) on the MedQA dataset with the LIME explanation method, which we believe is a reasonable approximation of real-world setups given the popularity of feature attributions in medical settings [1]. Our findings show that a calibrated GPT-4o-mini places greater importance on clinically relevant words, as measured by LIME.
>
>
> **W2:** We have updated Appendix A2 with additional ablation experiments and discussions on the training dynamics. Aligning with general ML intuition and calibration-specific results [2], we find that overfitting tends to happen when the training set size is below or near the number of parameters, and that increasing it reduces this risk. In our setting, the number of parameters scales quadratically with the number of classes (specifically, $m^2 + m$ as MCal is parametrized by a single `nn.Linear` layer), which in all of our experiments was only $m \leq 8$. With few exceptions (e.g., PhysioNet with a train/test split of 840/210), most of our training datasets have larger magnitudes, and we thus have reasonable confidence that overfitting did not occur in our experiments.
>
>
> **Q2:**  There are two general strategies to prevent overfitting when the class space is large. The first is to add a regularizer to the calibrator objective. Another is to consider a sparse parametrization of the calibrator, such as using a diagonal matrix for $W$ instead of a dense square matrix. This is also known as “vector-scaling” in [2], with a parameter count of $O(m)$, and is known to offer a good performance-efficiency trade-off compared to the full matrix variant.
>
>
> **Q3:** Our experiments show that test-time accuracy improves with the training set size. The training time broadly scales with the dataset size, and all runs finished in $\leq 5$ seconds in part due to our simple and optimizer-friendly (convex) parametrization, even for our largest setup of 5000 samples and 5000 iterations. Timing variations can be attributed to the shared server we were using. Finally, while we were not under the resource pressure to balance dataset size and computational efficiency, this is a valid concern. In such cases, we recommend the use of vector-scaling parametrization as described above, while keeping the training set size as large as possible to avoid overfitting.
>
>
> [1] Chen et al. “Algorithmic fairness in artificial intelligence for medicine and healthcare”. Nature Biomedical Engineering, 2023.
>
>
> [2] Guo et al. “On Calibration of Modern Neural Networks”. ICML 2017.

---

### Official Review · Reviewer_3Csh · 2025-10-30

**Soundness:** 2
**Presentation:** 2
**Contribution:** 2
**Rating:** 4
**Confidence:** 4

**Summary:**

This paper introduces a straightforward plugin to enhance explanation quality, which is clearly described and compatible with perturbation-based feature attribution methods. The authors propose to append the to-be-explained model with an additional dense layer, and fine-tune the added layer with the expected manipulations on the input samples while keeping the major part of the model frozen. The proposed solution is intuition-guided and empirically evaluated under small-scale settings, and the experimental results have an emphasis on reporting model performances rather than the intended explanation quality improvement.

**Strengths:**

- This paper is well-motivated and self-contained, with a clear description of the proposed method.
- The described plugin is flexible and can be appended to arbitrary models regardless of input modalities or model architectures.
- The empirical results demonstrate the capability of MCal in aligning the predictions on manipulated inputs with the standard model outcomes, supporting the method design.

**Weaknesses:**

- The discussion on the central motivation is insufficient. Particularly, line 120 states that “masking non-critical regions …”, yet it is arguable that considering certain regions as “critical” already injects human inductive biases. If the model truly learned to look at a meaningless/wrong region, would this additional “correction” for the missingness bias consequently cover up its problematic behavior?
- While the objective of the paper is clearly stated, the line of discussion deviates from how the added component will enhance explanation quality, instead leaning towards the effectiveness in resolving missingness bias. The modification, which can be considered a form of data augmentation, is intended to facilitate the derivation of more faithful explanations; however, the designed experiments focus more on the impact of this augmentation on model stability and do not sufficiently demonstrate how the modification leads to better explanations.
- In fact, this paper consistently mixes the discussion on feature attributions with model robustness. Although related in some ways, they are different subjects, where the missingness bias is a particular challenge that feature attribution methods are facing, and robustness is an inherent property of a model. A properly designed explainer should be able to faithfully reflect model behaviors, no matter whether it is robust or not. Mixing these two topics loses the concentration on explainability, and particularly disconnects the latter part of the discussion in this paper from its stated motivation.
- The faithfulness of the derived explanations becomes questionable given the added dense layer. While adapting the tested model to the disturbed manifold, it arguably also changes the model behavior as a whole.
- Some experimental settings and result interpretations are questionable; see questions for more details.

**Questions:**

- See the first point in Weakness.
- Could the authors better explain the objective of this paper? What are the takeaways of the stability test on the calibrated models? How do they support the claim of improved explanation quality?
- Do the explanations generated on the calibrated model still faithfully reflect the behavior of the original model?
- Figure 5 presents the only results that are relevant to explanation quality. Could the authors elaborate on the interpretation of the sensitivity metrics reported there? Given the focus is on explanation quality, the sensitivity measure should reflect the effectiveness of an explainer in identifying the most relevant features. Excluding them should lead to a larger drop in prediction confidence (as stated in line 726); thus, a higher sensitivity score should indicate more effective explanations. In this sense, the results are in favor of explanations from uncalibrated models, which contradicts the interpretation presented in the paper. Also, I do not follow the argument related to model robustness and its connection to the evaluation of explanation quality.
- Could the authors clarify the different training settings for retrain and MCal? Particularly, why is the retraining scheme fitted to a different ablation distribution than MCal? MCal was fine-tuned exactly for the stratified mask sampling that is exactly used for the validation, which has a balanced distribution of ablation fractions, whereas the retraining scheme only sees the results of uniformly ablated inputs, lacking observations of rich and rare feature presences (e.g., ablation fractions of 2/16 and 14/16 are rarely possible to be sampled). I’m surprised and, in fact, skeptical about the conclusion due to the misaligned ablation strategies.

---

> ### Author Response · Authors · 2025-11-28
>
> We thank the reviewer for their patience, as well as for their insightful questions that have helped us hone our paper’s positioning. We respond to the main points below.
>
> **W1 + Q1 + Q2:** Thank you for pointing out this area for improvement. Our primary motivation is to mitigate the missingness bias phenomenon in feature attribution methods. Models with high bias are known to induce poor explanations [1], which makes them unreliable to deploy in safety-critical settings [2, 3]. We approach this by designing a calibrator to remove this bias while maintaining performance. We will update our manuscript to sharpen the motivation.
>
> To expand on missingness bias: Perturbation-based feature attribution algorithms operate under the core assumption that masking critical local features for the model, not necessarily aligned with human inductive biases, should result in lower confidence scores. We simply argue that the masks themselves result in undesirable, unaccounted-for behavior, which can affect explanation quality, building upon works such as [1]. A demonstrative example is in [1], where black masks at irrelevant locations can lead to misclassification as a jigsaw puzzle. This is a violation of the assumptions of feature attribution algorithms and the problem we chose to tackle.
>
>
> **W2 + Q2 + Q4:** Thank you for raising this point and helping us sharpen our analysis. Based on these discussions, we have now split the sufficiency and sensitivity experiments (we assume you meant these by “stability”) as explanation quality (Question 1) and model robustness (Question 2) in the revised manuscript.
>
> The key signal for improved explanations is in the sufficiency score. A lower sufficiency score is evidence for higher explanation quality, as also referenced in [2], and measures the confidence drop when only top-$k$ features are retained:
>
> $$
> \\text{Sufficiency} = f(x)_y - f(\\text{KeepOnly}(x, \\text{top-}k))_y
> $$
>
> We see in Figure 5 exactly this behavior: a lower sufficiency score on the calibrated model suggests that the explainer’s top-k ranked features capture a greater portion of the original confidence.
>
>
> **W4 + Q3:** The reviewer is correct: any explanation derived with respect to the calibrated model is faithful to the calibrated model only, not the base model. One intended use case is that the calibrated model, not the biased base model, is to be deployed in contexts where feature attribution methods are also used. Because our experiments suggest that the calibrated model is competitive with the base model, we envision that there exist scenarios in which this substitution is acceptable.
>
>
> **Q5:** There are two settings for MCal training: conditioned mode and unconditioned mode. In conditioned mode, which is what is reported in Table 1, MCal is fit separately on each fraction of missingness as the reviewer has correctly pointed out. In unconditioned mode, features are ablated with uniform probabilities, exactly resembling the nature of explainability algorithms such as LIME and SHAP, and this is the same distribution as observed by the model in the retraining scheme.  Notably, our experiments in Figure 6 show that unconditioned MCal is competitive with conditioned MCal, indicating that rich and rare feature presences may not play too much of a role in calibrator performance.
>
>
> [1] Jain et al. “Missingness Bias in Model Debugging”, ICLR 2022.
>
>
> [2] Hase et al., "The Out-of-Distribution Problem in Explainability and Search Methods for Feature Importance Explanations", NeurIPS 2021.
>
>
> [3] Duan et al., “On the Evaluation Consistency of Attribution-Based Explanations”. ECCV 2024.

---

### Official Review · Reviewer_Tchr · 2025-10-31

**Soundness:** 2
**Presentation:** 3
**Contribution:** 2
**Rating:** 6
**Confidence:** 3

**Summary:**

The paper proposes a calibration technique to mitigate the missingness bias problem encountered when finding feature importance scores. The technique proposed is lightweight and requires access only to logits. They evaluate their technique on various medical data across vision, language, and tabular modalities.

**Strengths:**

1) The proposed calibration technique is lightweight, not requiring retraining, access to weights, or architectural modifications of the original model.
2) The proposed technique can be well adapted to common perturbation-based explainability techniques.
3)The paper is generally well written, easy to follow, and the motivation is well presented.

**Weaknesses:**

1)In baseline comparisons, for the replace and retrain categories, the paper uses basic techniques instead of comparing against more advanced techniques in those categories. For example, in the retrain category, some of the recent techniques for imputation, like ROAD, GOAR, are not compared against. Since these are relevant methods to tackle the missingness bias, it would be better to compare them.
2)The paper claims the missingness bias is a superficial artifact, while the original model embeddings might still be facing this issue.
3)Some of the crucial details of the experiments are missing. For Figure 7, it isn’t clear if the accuracy is computed using ablated or clean input.
4)The details of input ablation rates for the baselines in Table 1 are not mentioned, while the proposed method reports an average of calibrators conditioned on various ablation rates.

**Questions:**

Based on the argument in the paper that missingness bias can cause class distributional shift, couldn’t the ablation of seemingly unrelated or spurious features, which the model might rely o,n contribute to this distributional shift?

---

> ### Comment · Reviewer_Tchr · 2025-11-25
> **No response received**
>
> The authors have not addressed the questions/comments provided earlier. Hence, the rating remains unchanged.

---

> ### Author Response · Authors · 2025-11-28
>
> We thank the reviewer for their patience, as well as for their positive assessment of the paper and helpful feedback. We address the main points below.
>
> **W1:** Thank you for these suggestions. Indeed, more modern imputation and retraining approaches may reduce missingness bias and outperform MCal. Unfortunately, neither ROAD nor GOAR is explicitly designed to mitigate missingness bias: ROAD is intended to protect against information leakage in feature attributions, whereas GOAR is designed to measure the fidelity of feature attributions. We chose to include them as references because they address adjacent explainability problems, and we have clarified the wording around this.
>
> Nevertheless, the methods in ROAD would be a good baseline, and we will work on adding this in future revisions of the manuscript, given that it is a surprisingly engineering-intensive endeavor. On the other hand, it is non-trivial to directly compare with GOAR, as its retraining scheme depends on a fixed choice of feature attribution method. However, exploring how the ideas therein could be extended to be agnostic to this choice would be an interesting research direction.
>
>
> **W2:** Our choice of wording reflects our observation that a linear correction to model logits suffices to mitigate missingness bias. This suggests that even though the model’s embeddings may be biased, this bias may not be so severe in the context of feature ablations. We have improved our discussion on this.
>
>
> **W3:** Thank you for pointing these out. We have expanded our experiment details (also see Appendix A). In Figure 7 of the submission version, the accuracy is computed with respect to both the clean and ablated inputs, where an ablation fraction of 0 indicates a clean input.
>
>
> **W4:** The ablation rates are described in the “Input Ablations and Calibration” paragraph of the submission version. In particular, we use $p \in \\{0/16, 1/16, \ldots, 15/16\\}$ for image data, and $p \in \\{0/10, 1/10, \ldots, 9/10\\}$ for both language and tabular data.
>
>
> **Q1:** The reviewer’s understanding is correct: the ablation of spurious features indeed contributes to the distribution shift that induces missingness bias. This is exactly the type of bias that we hope to address with MCal.

---

### Official Review · Reviewer_7kkY · 2025-11-01

**Soundness:** 2
**Presentation:** 2
**Contribution:** 2
**Rating:** 4
**Confidence:** 4

**Summary:**

The objective of this paper is to enhance explanation quality by resolving the missingness bias, which is a consequence of distribution shifts caused by input manipulations during the explanation procedure. The motivation is clearly stated, and the proposed solution, while intuitive, is reasonable. However, the discussion throughout the paper consistently mixes the pursuit of better explanation quality with model robustness, which are two distinct topics.

**Strengths:**

- The proposed method is straightforward and well-presented, but the analyses and discussions on its impacts lack depth.

**Weaknesses:**

- In fact, if taking a data augmentation perspective, the proposal of this work is not brand new.

- While the experiments focus on elaborating how MCal corrects the missingness bias, the results fail to show how the calibration on model outcomes affects explanation quality. There is no qualitative example that illustrates the differences, nor quantitative assessments following standard regimes that effectively compare explainers under different settings.

**Questions:**

- The validity of the conclusion from the observations is somewhat questionable. For example, Figure 5 claims that “calibrated models have better explanations”; however, two versions generally show the same perturbation tendencies. The differences in magnitude appear to be more relevant to the changed model behavior due to the additional dense layer. Additionally, it is noteworthy that the two tested models have different functionalities due to the appended dense layer in one version. It is unclear how the results from different explanations on different models can be effectively compared.

- The results presented by the middle chart of Figure 7 appear suspicious. Why is the accuracy a constant with increasing ablation fraction for Conditioned MCal? An ablation rate of up to 90% is very likely to remove all informative features in an input. How does the model manage to maintain the accuracy without “seeing” anything relevant?

- This is more about a question that is relevant to the previous point. The example in Figure 2 motivates a correction of model outcomes when the relevant region is not masked, but what should be the target label if all relevant regions are masked out? Enforcing the output of disease on a manipulated input without indications is arguably another form of “skewed” prediction, particularly when the model originally predicts healthy correctly. I think the argument of this paper only builds upon one side of the coin and leaves the other side insufficiently discussed, and the relevant difficulty unsolved.

---

> ### Author Response · Authors · 2025-11-28
>
> We thank the reviewer for their patience and constructive feedback, which have helped us improve the manuscript. We respond to the main points below.
>
> **W1:** We wish to clarify that a key novelty of our work is in using calibration to mitigate missingness bias. The fact that calibration on ablated inputs can effectively address missingness bias is impactful because it suggests that a longstanding problem in explainability can be significantly mitigated via a simple method.
>
>
> **W2:** Experiment 1 (Figure 5, Appendix A1) is intended to show that MCal improves the explanation quality, and we expand on our rationale in the Q1 response.
>
> Moreover, we refer the reviewer to our newly updated Appendix A3 for a case study of applying MCal to API-based models. There, we apply MCal to GPT-4o-mini (via the logits API) on the MedQA dataset, in particular finding that the calibrated logits place greater importance on clinically relevant words, as measured by LIME.
>
> Furthermore, our new Appendix B contains visual examples of LIME-generated explanations for the MRI dataset, both for calibrated and uncalibrated models. We observe that calibrated models place less weight on border patches, where brain tumors are less likely to occur.
>
>
> **Q1:** These are great points. We format them into several categories below.
>
> * **Explanations on calibrated models:** Thank you for this sharp observation: indeed, lower sensitivity is a reflection of model robustness, and we have now decoupled this to a separate experiments section. However, a lower sufficiency score is evidence for higher explanation quality, where recall that it measures the confidence drop when only top-$k$ features are retained:
>
>
>
> $$
>  \\text{Sufficiency} = f(x)_{y} - f(\\text{KeepOnly}(x, \\text{top-}k))_y
> $$
>
>
>
> A lower score thus means that the top-k ranked features by the explainer capture a greater proportion of the original confidence, which is affirmed by Figure 5’s sufficiency plots.
>
>
> * **Same perturbation tendencies:** We agree that both models share similar tendencies, as performance degradation under ablation is expected. However, the critical differentiator is the magnitude of this degradation. The consistent vertical gap in Figure 5 demonstrates that the calibrated model suffers significantly less from missingness bias.
>
>
> * **Inclusion of dense calibrator layer:** We clarify that the added component is a strictly linear transformation ($Wz + b$), not a generic dense layer with non-linearities. Because the base model already ends with a linear classification head, and the composition of two linear layers is equivalent to a single linear layer, adding MCal does not expand the model's representational capacity. That is, although the calibrated model has lower missingness bias, it is not more expressive than the base model.
>
>
> **Q2:** Thank you for pointing this out. In fact, this was due to a plotting error. We have rerun the experiment and now observe the expected trend in accuracy preservation, and updated our plots.
>
>
> **Q3:** For MCal’s objective, the target label for ablated inputs remains the original class prediction from the clean input. While one would ideally like to have an oracle (or human) re-label the ablated input, this is often not available. Consequently, we take the prediction on the clean input as a best guess. While this approximation risks incorrectly labeling data (e.g., all tumorous regions in the MRI masked), our experience suggests that this is rare. In general, we observe that salient information is often spread across multiple features, meaning that random ablations are unlikely to simultaneously mask all the features. In fact, if this were not the case, then we would have observed poor accuracy in the calibrated model (see Figure 7).

---

### Author Response · Authors · 2025-12-04
**Rebuttal Summary**

We sincerely thank all reviewers and Area Chairs for their time and thoughtful evaluation. We have addressed the main concerns of all reviewers and uploaded a revised manuscript. A summary of the rebuttal is provided below.


**(Reviewer 7kkY)** The main concerns were on the novelty of our work and the distinction between quality and robustness in explanations. To that end, we have added: Clarification that MCal's novelty lies in using calibration for missingness bias mitigation (W1), additional qualitative examples in Appendix B and an API-based LLM case study in Appendix A3 demonstrating explanation quality improvements (W2), clarifications to Figure 5 including the distinction between sufficiency and sensitivity metrics (Q1), correction of the plotting in Figure 7 (Q2), and discussion of the label approximation strategy for heavily ablated inputs (Q3).


**(Reviewer Tchr)** The main concerns were on the choice of baselines, wording of “superficial artifact”, and experiment details. For this, we have improved: Discussion of ROAD/GOAR as adjacent methods rather than direct baselines with plans for future comparison (W1), clarification that "superficial artifact" reflects our finding that linear correction suffices (W2), expanded experiment details in Appendix A regarding accuracy computation (W3), and clarification of ablation rate specifications (W4).


**(Reviewer 3Csh)** The main concerns were the motivation of our work and the clarification of experimental details, as well as the interpretation of our results. We have: Sharpened motivation around missingness bias and its impact on explanation reliability in safety-critical settings (W1, Q1, Q2), separation of sufficiency (explanation quality) and sensitivity (model robustness) experiments to address the concern about mixing topics and interpretation of experiments (W2, W3), clarification that calibrated models are intended for deployment alongside feature attribution methods (W4, Q3), detailed explanation of sufficiency metric interpretation (Q4), and clarification of conditioned vs. unconditioned MCal training settings showing comparable performance to clarify confusion regarding algorithmic details and the sound interpretation of our results (Q5).


**(Reviewer ojnE)**  The main concerns were on the extension of MCal to real-world case studies and analysis of training dynamics. We have added: A new case study in Appendix A3 applying MCal to API-based LLMs as a real-world application scenario with closed-weight models (W1, Q1), additional ablation experiments on training dynamics in Appendix A2 (W2), assuaging concerns of overfitting and discussion solutions for scenarios in which it may happen (Q2), and analysis of training set size effects on performance with timing details (Q3).


**(Reviewer 3ZQX)** The main concerns were the impact of missingness bias across different domains and ways to mitigate it, e.g., using API-based models. We have added: Discussion of missingness bias prevalence across natural image datasets (W1), new qualitative visualizations in Appendix B comparing saliency maps on calibrated vs. uncalibrated models (W2), and implementation of an API-based LLM case study with GPT-4o-mini to demonstrate adaptiveness to modern machine learning paradigms and use-cases (W3).


Because the discussion process was frozen before there was time for the reviewers to engage with our rebuttal, we hope that the above summary will be helpful under this new process.

---

### Meta-Review · Area_Chair_5DkZ · 2026-01-06

**Summary:**

This paper introduces MCal, a lightweight post-hoc method to address "missingness bias" in feature attribution explanations. Missingness bias occurs when models are probed with ablated, out-of-distribution inputs during explanation procedures, leading to unreliable feature importance scores. The authors propose that this bias can be treated as a superficial artifact correctable through fine-tuning a simple linear head on the outputs of a frozen base model rather than requiring expensive retraining or architectural modifications, and also offering theoretical guarantees of convergence for the calibration method, providing confidence in reproducibility and stability. While the reviewers consistently praised the practical nature of the proposed solution, several reviewers questioned the novelty of the approach, suggesting it resembles data augmentation techniques. Three reviewers leaned positive while two leaned negative, though all indicated they would not strongly object to the alternative outcome. The authors provided substantive responses addressing the major concerns, including new case studies, visualizations, and clarifications distinguishing explanation quality from model robustness. However, Reviewer Tchr noted that their concerns were not addressed, and the authors acknowledged that the discussion process was frozen before reviewers could fully engage with the rebuttal. Given the practical value of the proposed lightweight solution, the comprehensive revisions made, and the borderline positive aggregate sentiment, I recommend accepting the paper while encouraging the authors to incorporate the promised comparisons with ROAD/GOAR and expand real-world evaluations in the final version.

**Reviewer Concerns:**

Reviewers requested real-world case studies and evaluation on API-based models to demonstrate practical applicability. In response, the authors added a case study in Appendix A3 applying MCal to GPT-4o-mini on the MedQA dataset. They also added qualitative visualizations in Appendix B comparing saliency maps on calibrated versus uncalibrated models.

**Reviewer Scores:**

Given the rebuttal results, one of the Reviewers 7kkY and 3Csh (Rating 4) could have changed their score to 6.

---

### Decision · Program_Chairs · 2026-01-26

Accept (Poster)